# Linking climate warming and land conversion to species' range changes across Great Britain

Andrew J. Suggitt [1] ✉, Christopher J. Wheatley[2], Paula Aucott [3], Colin M. Beale [4,5], Richard Fox [6], Jane K. Hill [2], Nick J. B. Isaac [7], Blaise Martay[8], Humphrey Southall[3], Chris D. Thomas [2], Kevin J. Walker[9] & Alistair G. Auffret [10] ✉

Although increased temperatures are known to reinforce the effects of habitat destruction at local to landscape scales, evidence of their additive or interactive effects is limited, particularly over larger spatial extents and longer timescales. To address these deficiencies, we created a dataset of land-use changes over 75 years, documenting the loss of over half (>3000 km²) the semi-natural grassland of Great Britain. Pairing this dataset with climate change data, we tested for relationships to distribution changes in birds, butterflies, macromoths, and plants (n = 1192 species total). We show that individual or additive effects of climate warming and land conversion unambiguously increased persistence probability for 40% of species, and decreased it for 12%, and these effects were reflected in both range contractions and expansions. Interactive effects were relatively rare, being detected in less than 1 in 5 species, and their overall effect on extinction risk was often weak. Such individualistic responses emphasise the importance of including species-level information in policies targeting biodiversity and climate adaptation.

Biological assemblages are changing rapidly, and although changes in land use and climate are widely considered to be the foremost drivers of these shifts[1], our understanding of their additive and interactive effects over multi-decadal time periods is limited. This is often due to a mismatch between biological and environmental datasets, and so investigations are forced to rely on space-for-time substitution[2], or infer biodiversity change using species-area relationships[3]. Climate impact studies often use present-day configurations of land use as a surrogate for habitat availability, yet we already know that land use has

changed over the same period as warming has occurred[4]. An inability to disentangle these two key drivers of biodiversity change has obvious implications for policy and practice designed to ameliorate them, as interventions that target one particular effect may not be effective in the presence of both.

Land-use change can affect the capacity of species to respond to climate change, and can impact the magnitude of local and regional climate change, accelerating or dampening the overall pattern of warming in different parts of the globe[5]. Climate change, and

[1]Department of Geography & Environmental Sciences, Northumbria University, Newcastle upon Tyne NE1 8ST, UK. [2]Leverhulme Centre for Anthropocene Biodiversity, Department of Biology, University of York, York YO10 5DD, UK. [3]School of the Environment, Geography and Geosciences, University of Portsmouth, Portsmouth PO1 3HE, UK. [4]Department of Biology, University of York, York YO10 5DD, UK. [5]York Environment Sustainability Institute, University of York, York YO10 5DD, UK. [6]Butterfly Conservation, Manor Yard, East Lulworth, Wareham, Dorset BH20 5QP, UK. [7]Centre for Ecology and Hydrology, Maclean Building, Benson Lane, Crowmarsh Gifford, Wallingford, Oxfordshire OX10 8BB, UK. [8]British Trust for Ornithology, Beta Centre (Unit 15), Stirling University Innovation Park, Stirling FK9 4NF, UK. [9]Botanical Society of Britain and Ireland, Room 14, Bridge House, 1-2 Station Bridge, Harrogate, North Yorkshire HG1 1SS, UK. [10]Department of Ecology, Swedish University of Agricultural Sciences, 75007 Uppsala, Sweden. ✉e-mail: andrew.suggitt@northumbria.ac.uk; alistair.auffret@slu.se

particularly trends towards warmer or drier climates, can also drive changes in land cover[6], and by extension, land uses associated with that cover. These phenomena are known to have to combined[7] or interacted[8,9] to define biological responses to environmental change. However, a lack of spatial and temporal coverage of gridded datasets quantifying these drivers – particularly so in the case of land-use change – has limited efforts to simultaneously establish both their prevalence and their long-term effects (>50 years) across a range of taxa.

In this work we focus on Great Britain, where the biota has been subject to substantial changes in land-use and climate, and where biological recording extends sufficiently far back in time to study long-term effects[10–12]. Using a large dataset of over 20 million species' distribution records, we show that interactive effects of land-use change and climate change are relatively rare, and that these two factors instead tend to act individually or additively to drive distribution change. Our findings further highlight the individualistic responses of species to environmental change[13], re-emphasising the need for interventions targeting biodiversity to be devised and delivered at the species level.

## Results

### Mapping long-term land-use change in Great Britain

Although climate datasets for Great Britain have good spatial and temporal coverage, no comparable dataset exists for land-use change prior to the satellite era (although analyses have been conducted for an English county[14]). Here, we digitised historical maps[15] of land use from the 1930s and 1940s (Fig. 1a) to derive the first such map for the whole of Great Britain. Obtaining a recent satellite-derived survey of land cover[16] (Fig. 1b), and harmonising the native land use classes across the two maps (Table S1), we quantified land-use change in Great Britain over the last 75 years (Fig. 1c–i) at the 10 × 10 km grid square level. We identified substantial changes in land-use (Fig. 1c, d), including increases in arable land (from 22% to 27% of total land cover), urban (from 4% to 5%), woodland (from 6% to 12%), and agriculturally-improved grassland, the latter expanding from trace levels[17] to occupy some 24% of the land area (Fig. 1i). These changes were largely at the expense of semi-natural grassland (including heathlands, as well as lowland meadow and pasture), which declined from 65% to 30% (Fig. 1h). Most significantly, semi-natural lowland meadow and pasture, an ecologically important subdivision of the semi-natural grassland category, shrank by 90% over the 75 year time period (Fig. 1d). Overall, rates of both climatic warming and land conversion exhibited substantial spatial variation within Great Britain (Fig. 2a, b) to which species may have responded.

### Linking land-use and climate change to species' distributions

We used the new land-use change dataset and the existing climate change dataset to look for single, combined, and interactive effects of temperature warming (Fig. 2a) and land conversion (which we define as the proportion of 25 × 25 m pixels within a 10 × 10 km grid square that had changed land-use class over time, Fig. 2b) on species'

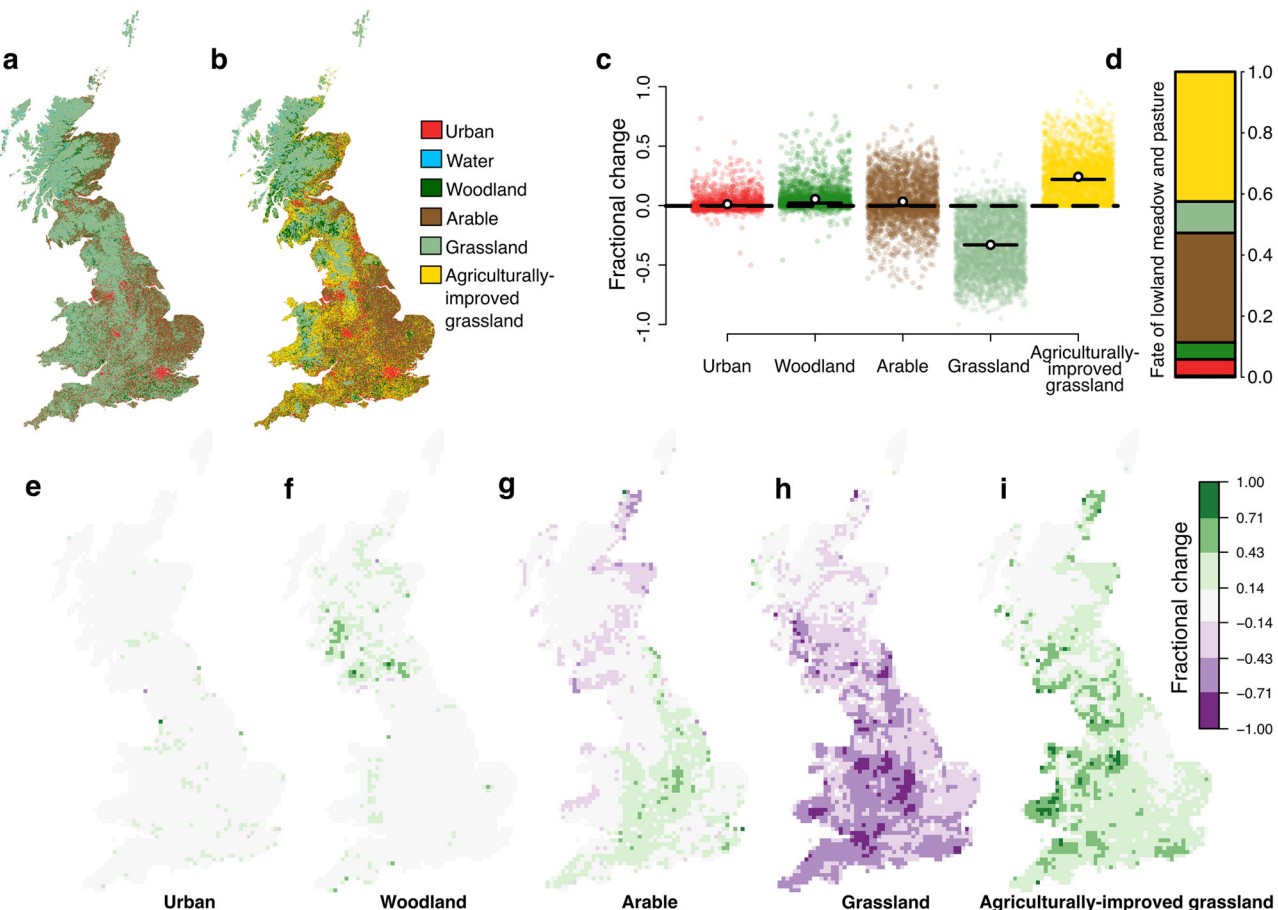

**Fig. 1 | Land-use change in Great Britain over the last 75 years. a** Land use was mapped for the historical[13] time period (LUSGB 1930–40 s, copyright Giles N. Clark) and (**b**) the modern[14] time period (LCM 2007) at 25 m × 25 m resolution in broad land-use categories. **c** Fractional change (−1 ≤ x ≤ +1) in the five terrestrial land use categories between the historical and modern time period. Each coloured point represents a 10 × 10 km grid square, white circles show mean values and horizontal lines represent median values (across all grid squares). **d** The fate of pixels originally classified as lowland meadow and pasture, with transitions to the broad land use categories illustrated proportionally. **e–i** Fractional change in coverage of the five terrestrial land use categories within grid squares.

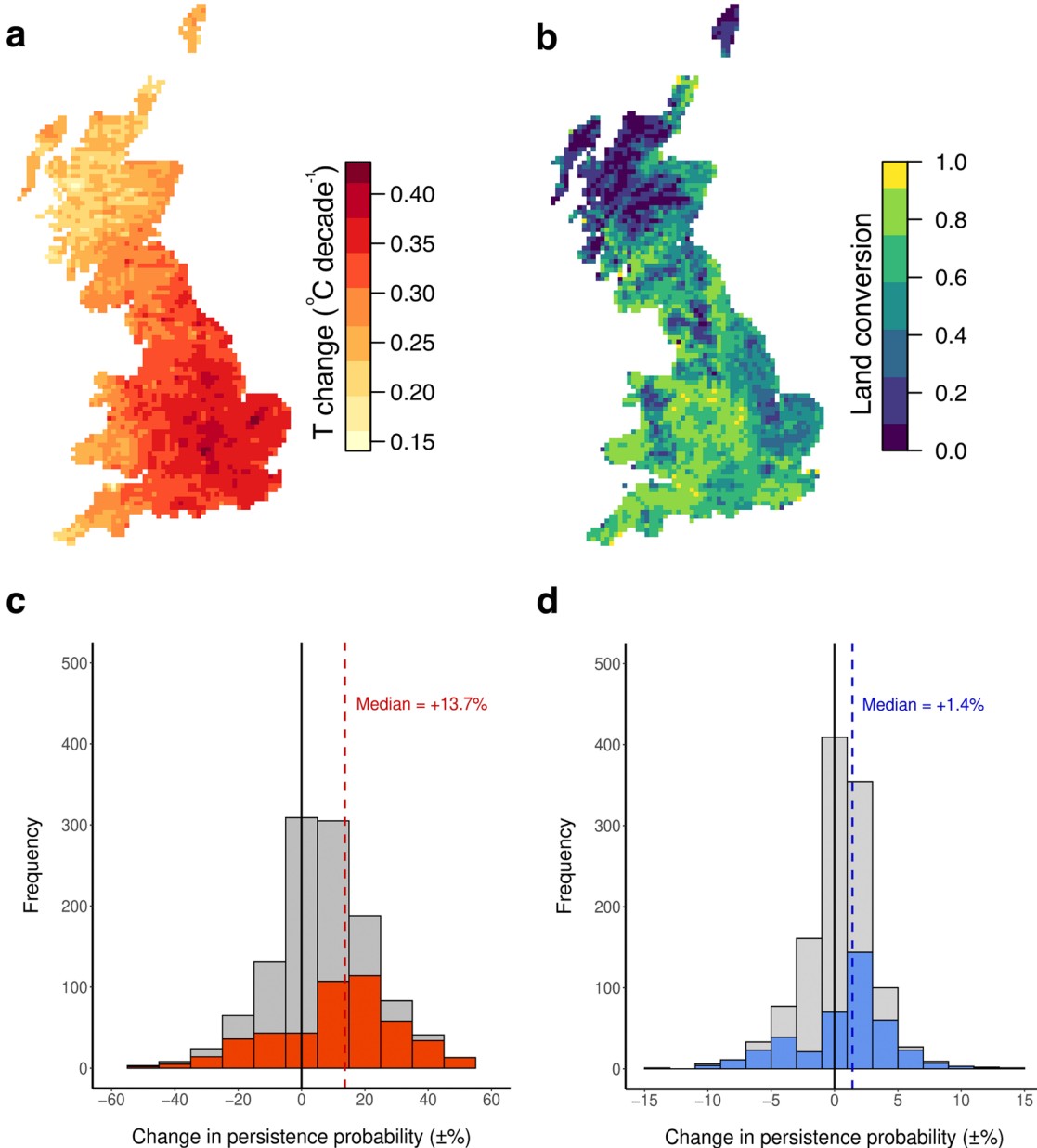

**Fig. 2 | Distributional responses to long-term changes in land use and climate.**
**a** The rate of mean temperature (T) change was calculated for Great Britain, at
$10 \times 10$ km grid square resolution. **b** a 'land conversion' map was also generated,
where values indicate the proportion of $25 \times 25$ m pixels within each $10 \times 10$ km
square that changed category between the two time periods, interval -75 years.
Long-term persistences or extirpations for 1192 species were modelled in response
to various combinations of land-use change, temperature change, and their inter-
action. Models fitted to the data for each taxon included both temperature change

and land conversion, suggesting a positive effect of each variable on persistence
(with an antagonistic interaction, in the case of birds), but a high proportion of these
models' explanatory power was accounted for by species-level random effects.
Subsequent models fitted to data for each species revealed large variation in the
Average Marginal Effects (AMEs) of (**c**) a 0.1 °C per decade temperature rise and a
10% land conversion (**d**). All 1192 species are illustrated in grey, with coloured bars
representing the subsets of species for which the respective variable was included in
the 'best' model (excluding species exhibiting interactive responses, $n = 230$).

distribution changes. We assembled over 20 million biological
records to build a database of distributions at the $10 \times 10$ km grid
square level, covering the whole of Great Britain. Our database
covered 1192 species of bird, butterfly, macromoth, and plant. We
mapped the distributions of these species in two time periods: a
1930–1972 'historical' and a 2000–2015 'modern' period, based on
the availability of records for each taxonomic group (Table S2). To
ensure our statistical analyses were robust, we excluded any species
that was recorded in 100 unique $10 \times 10$ km grid squares or fewer in
the first time period ($n$ grid squares in Great Britain = 2911).

We considered persistence (presence in a grid square in both time
periods) versus extirpation (presence in the first period but absence in

the second) for each species over time. We modelled these outcomes
statistically, fitting logistic linear mixed-effects models (LMMs) to the
data on population outcome (1 = persisted, 0 = extirpated), with land
conversion (fraction of grid square where land-use category changed
over time), temperature change and their interaction as predictor
variables, and species treated as a random effect. Additional variables
controlled for microclimatic buffering[18], recorder effort[19] and spatial
autocorrelation[20]. We tested five different model formulations along a
gradient of complexity (Table S3), from a null model including only the
control variables, to a full model including controls and land conver-
sion, temperature change and their interaction, identifying the best
performing model with the lowest AIC[21]. We interpret statistical

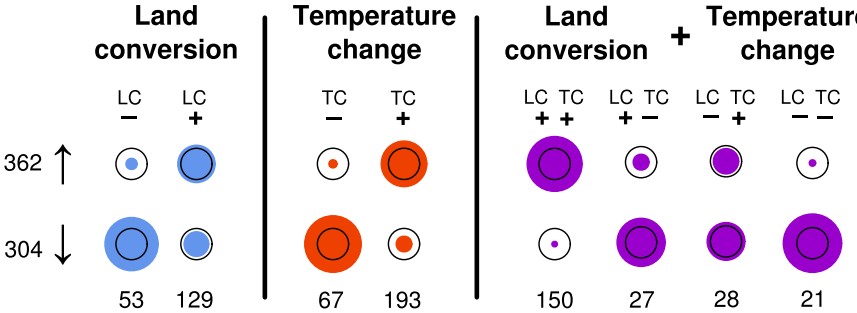

**Fig. 3 | Translation of grid-square persistence into distribution trends across Great Britain.** Species are sorted into columns based on the identity of their 'best' model (n = 668 species), with total numbers of species provided in the bottom row. The size of the coloured circles is proportionate to how many species had positive (overall n = 362) or negative (n = 304) distribution trends. Two plant species experienced zero distribution change and were excluded from this analysis. Black circles indicate the radii of what would be an even split of species with positive and negative trends in each column; thus, a coloured circle falling within this radius indicates that fewer species were associated with a particular response cohort than expected, and a coloured circle exceeding this radius indicates that more species were associated with a particular response cohort than expected.

associations to environmental change variables as evidence of effects, and for simplicity we use terms such as 'effects' and 'drivers' hereafter. As a first step, we constructed models by grouping the data for each taxon together, and treating species identity as a random effect, but in all four taxa the proportion of variation in persistence explained by this effect in these models was high (Table S4), so we proceeded by fitting models at the species level.

We fitted equivalent models to the persistence/extirpation data for each species using Generalised Linear Modelling (GLM) approaches, again varying model complexity and selecting the 'best' model for each species via AIC. As with the previous analysis, the need to include changes in both climate and land use in analyses of range change was apparent. The best performing models for 75% of analysed species (n = 898) included land conversion, temperature change, or both, leaving some 25% of species (n = 294) for which the controls-only model was best (Fig. S1). For some 56% (n = 668) of analysed species, the best-performing model included environmental change variables without an interaction term, and of these species, some 472 (or 40% of the total) unambiguously benefitted from environmental change- via their positive responses to land conversion, temperature change or both- while 141 (12% of total) species were unambiguously worse off. Across these 668 species exhibiting non-interactive responses, there was also substantial variation in the magnitude of the effect of environmental change on population persistence, as calculated using Average Marginal Effect (AME; Fig. 2c, d). This was particularly true for land conversion, where two modes of response were prevalent: a larger cohort of weakly positive responses to land conversion, and a smaller cohort of strongly negative responses (Fig. 2d). Considering only those species where the best performing model included the relevant environmental change variable (without an interaction term), persistence probability increased by a median of +13.6% for every 0.1 °C per decade increase in temperature experienced, and by +1.4% for every 10% of land converted within a grid square (Fig. 2c, d).

Interaction effects between land conversion and temperature change were included in the best-performing model for 19.3% of analysed species (n = 230). To assess if these interactions were meaningful, we estimated persistence probability across the range of observed changes in temperature change and land conversion using models formulated both with and without the interaction term, i.e., a 'full' model including the interaction term and an otherwise identical model that omitted the interaction term (additive model). We found that predicted estimates derived from the interactive model were different to those derived from the additive model for 30 species (Paired t tests, d.f. = 118, Bonferroni-corrected p-value = 0.0002) of the 230 species for which an interaction term improved the model. Differences in

persistence probability estimates across the two models remained below 10% across half of the observed parameter space of environmental change for some 197 out of 230 species (Table S5), with larger differences (of 40% or more) limited to 18–24 species and occurring in 30% of the parameter space (at most, Table S5). We therefore conclude that although there are large numbers of species for which combined effects of temperature change and land conversion are driving range changes (here 38% of all considered species), meaningful interactions between land use change and climate change on species persistence are a relatively rare phenomenon affecting a minority of species (here ~30 of 1192 analysed species, or 2.5% of the total).

To determine the extent to which habitat specialists were more sensitive to changes in land use (or land use-climate interactions) than the wider species pool, we conducted an additional analysis on a subset of species (n = 168 species) we identified as specialised to a single habitat within the European Red Lists for species[22]. We re-ran the statistical models of persistence vs. extirpation formulated as before, but this time substituting the generic 'land conversion' variable for relative change in the specific habitat type each specialist was associated with. Responses of specialists were broadly similar to the full species pool (Fig. S3 vs. Fig. S1); however, for those species with habitat change in best performing models (without interactions), the AME of habitat change was larger (median 1.8% vs. 1.4% per 10% increase in cover of specialist habitat type, Fig. S4). Grassland specialists appeared more sensitive, with a + 10% change in semi-natural grassland habitat (more a case of retention rather than increase; Fig. 1h) resulting in a median 1.95% increase in the persistence probability. Increases in woodland cover also had a positive, albeit less strong, effect on persistence probability in woodland specialists (AME median +0.79% per 10% increase in cover). For the 30 species where an interaction effect between habitat change and climate change was included in the best model, predictions derived from that model differed from the analogous additive model (omitting an interaction term) for 5 species (Paired t tests, d.f. = 118, Bonferroni-corrected p-value = 0.0016), again a similar proportion to results from the full species pool.

Finally, a further analysis utilising a metric of observer-effort corrected distribution change[19] showed that where the best-performing model for a species contained only one of our main predictors, positive and negative species responses to climate in terms of grid-square level persistence generally translated into national level distribution gains and losses, respectively. For species responding to both variables, additive effects were clear, with a positive response to both temperature and land use made a positive distribution change much more likely, and vice versa (Chi-squared test: χ-square = 53.15, d.f. = 7, p < 0.0001; Fig. 3).

## Discussion

We show that the long-term persistence or extirpation of a species is often associated with the individual or additive effects of changes in climate and land use. These results emphasise that species' responses to environmental change are highly individualistic, and further illustrate how neutral biodiversity trends can in fact hide declines in many species. This was especially evident for species responding to additive (i.e., non-interactive) effects of temperature change and land conversion, as we found that negative responses to either or both environmental drivers were associated with higher likelihood of range decline at the national level (Fig. 3). It is therefore likely that at least some of the apparently individualistic shifts in species' distributions can be attributed to the species-level variation in responses to changes in climate or land use at smaller spatial scales. Taken in sum, our results are compatible with the large variation in species-level trends in occupancy and abundance over time[23,24]. Interestingly, although land conversion and temperature change often acted on species persistence in the same direction (i.e., + + or − −; Fig. S1), meaningful interactive effects were found to be relatively rare (Fig. S1, Table S6). This perhaps surprising result is not unprecedented in multispecies studies (e.g., in pollinators[25]), and implies that the mechanisms by which the changed conditions affect species, populations, and individuals are different, or at least, are not susceptible to compounding or accelerating effects of multiple drivers acting at once.

Our results demonstrate that the inclusion of information on historical land use substantially improves our ability to explain range shifting under climate change. Our new dataset quantified substantial changes in the land use of Great Britain over 75 years, where agricultural land has expanded at the expense of semi-natural grassland, echoing findings from elsewhere in Europe[3] and beyond[26]. We showed that habitat losses have had a negative effect on species' persistence for habitat specialists (Figs. S3, S4), but in the case of our broader species pool (including generalists), species varied widely in their response, and the balance of responses to land conversion was weakly positive. This may reflect: (a) the relative lack of habitat specialists amongst our study species (or study region) and/or the extirpation of specialist species prior to our study period, or (b) biotic mechanisms that support population persistence (if not abundance) in environments that are disturbed[27], heterogeneous[28], or in dynamic equilibrium[29]. In either case, we would emphasise that environmental change is unlikely to be beneficial to species' persistence per se, and indeed, space-for-time approaches conducted at a global level suggest that interactive effects of climate and intensive agriculture may drive widespread declines in biodiversity[9], particularly in the tropics- where it is likely that species will not be as adapted to disturbance from anthropogenic land use as they are in temperate Western Europe. Management changes within land use categories (as such beyond our scope here), such as agricultural intensification, have also been shown to be detrimental in recent decades (e.g., for birds[30]), and could surpass land conversion as stronger drivers of change in regions (such as Great Britain) where land-use configurations are now largely stable.

The potential discrepancy between our findings of overall weakly positive responses to environmental change and those of others[2,9] might also relate to our focus on range retractions, as opposed to overall changes in geographic range and/or abundance. This is because the processes that control the survival of populations and those that control other population characteristics (dispersal, density, competition, etc) may differ. For example, a relatively cool-adapted plant species reaching its warm (equatorward) range limit in our study area may be simultaneously: (a) declining in abundance throughout most of its range; (b) be able to (at least temporarily) persist in the majority of its historical range; and c) be temporarily alleviated from interspecific competition from warm-adapted species that have thus far been unable to colonise (due to high rates of isotherm shifts), whilst benefitting from the improved conditions for growth and/or survival that

warmer temperatures would bring. The latter is especially relevant for cooler regions in the mid- to high-latitude temperate zone, and as such is less likely to apply in regions where many species are already facing their upper thermal tolerance limit. We would also point out that, unlike global patterns of higher temperature change in cooler regions, Great Britain has in fact tended to experience faster rates of warming in areas with warmer mean temperatures (Pearson's $r = 0.56$, shared variance of 31%; see Methods). This could mean that a component of the positive responses to climate change may in fact be positive responses to higher mean temperatures in the environment. This positive correlation also makes it less likely that our approach has mischaracterised species' responses to (warmer/cooler) thermal environments overall.

Climate change is increasingly being shown to be an important driver of biodiversity change[31,32]. Here, we show that it might already have become both a primary threat to, and facilitator of, species persistence. Climate change effects are likely to strengthen over time, and an inability to colonise new areas could continue to hinder positive responses to climate[4,33]. Yet importantly, colonisation success also depends on the availability of suitable habitat within the landscape, and the thermodynamic properties of that habitat control its buffering potential for retracting species[8]- and both could act to increase the proportion of species affected by climate and land use effects simultaneously. Despite the broad taxonomic coverage of our study, it should be noted that rare or scarce species are often recorded less frequently, and were thus less likely to be represented here. We also did not consider other important components of the overall status of populations, such as abundance or colonisations. These factors, alongside the tendency for species' responses to global change to be highly individual, mean that the ultimate outcomes for many reorganised populations and ecosystems remain unclear.

## Methods
### Land-use change

The Land Utilisation Survey of Great Britain[15] (LUSGB) took place across the island of Great Britain during the 1930s and 1940s and generated land use maps at the 1:63,360 scale. LUSGB map sheets for England and Wales were scanned, georeferenced and combined into a raster mosaic[34], while the sheets covering Scotland were digitised individually (at a 5 × 5 m resolution) using the R[35] package *HistMapR*[36], which semi-automatically identified the respective map categories using their coloration. A number of the maps in rural Scotland were hand-coloured manuscript maps, where lakes and rivers were not adequately shaded, and were therefore difficult to separate from other land-use categories. For this reason, modern-day inland water from the Ordnance Survey Open-access vector map from 2016 was burned onto the digitised historical maps covering Scotland using the R package *gdalUtils*[37]. The LUSGB land use classes were as follows: [1] forest and woodland, [2] arable land, [3] lowland meadow and pasture, [4] heath and moorland, [5] suburban areas including gardens, orchards and allotments, [6] urban and industrial areas including roads, [7] inland water.

To test that the two different methods used to digitise the historical maps produced similar results, we compared land use in the 56 grid squares along the border region of England and Scotland, which were digitised using both methods. Overlapping pixels were assigned to the same land use category in 86% of pixels overall, with a root mean square deviation (i.e., landscape-level differences in relative land cover) of 4.5% at the 10 × 10 km grid square level, exceeding common targets for land-cover classification accuracy[38]. The historical maps were then merged into one dataset, keeping our own digitisations in these overlapping areas.

For a modern-day land use comparator, we used the 2007 Land Cover Map, which uses a largely (99.5%) automated method to classify satellite data to into 23 land cover classes[16]. To ensure accurate

comparison of these categories to those of LUSGB, we pooled the categories from both surveys into the following broad categories, using the scheme outlined in Table S1: [1] urban, suburban and otherwise built-up areas, [2] grassland (excluding so-called 'agriculturally-improved' grasslands; referred to in the main text as semi-natural grassland to better differentiate it from agriculturally-improved grassland), [3] arable land, [4] woodland, [5] inland water and [6] agriculturally-improved grasslands (present only at very low levels during the historical mapping period[14,17]). Littoral and sub-littoral land cover classes were removed because these areas were not consistently mapped in the historical maps, and were in fact often absent. Historical map digitisations were resampled to $25 \times 25$ m pixel size of the Land Cover Map using the *raster* package[39], and historical and modern map layers were masked against one another to ensure that they had the same extent. Because the historical maps were based on existing Ordnance Survey "Popular edition" maps (4th edition), roads were shown as being much bigger than they are in reality and were coloured in a similar (often identical) shade to urban areas. Therefore, to produce a usable and accurate land use map, we used the Ordnance Survey Open Roads dataset (as at March 2017) for Great Britain to remove all pixels from both the historical and the modern maps found within a 75 m buffer of the (modern) road network. A width of 75 m was selected as the best fit to the drawing width for roads in the original LUSGB surveys.

After producing the final land-use maps for each time period (Fig. 1a, b) at $25 \times 25$ m pixel resolution, the proportion of pixels each land-use category occupied in each $10 \times 10$ km grid square was calculated. Change over time for each land-use category within each $10 \times 10$ km grid square was calculated as the proportion in the modern maps minus the proportion in the historical maps (Fig. 2). Percentage cover at the national scale was also calculated (Fig. 1a, b), whilst we also quantified the proportion of $25 \times 25$ m pixels within a $10 \times 10$ km grid square that had changed land-use class over time (hereafter land conversion; Fig. 2b). Because the loss in semi-natural grassland is one of the most important drivers of biodiversity change in Europe (see red lists for e.g., birds[40], butterflies[41], and plants[42]), it is valuable to estimate the level of this loss in Great Britain during the 20th century. Unfortunately, overlapping definitions of rough grassland and heath, and lowland meadow and pasture across the maps of the two time periods meant that they had to be grouped together as grassland for the main analysis. This means that the generally high levels of retention of relatively species-poor heathland (e.g., areas of Scotland and Wales in Fig. 1) would lead to a large underestimation in species-rich grassland loss, even if it is not possible to know the land-use history of the lowland meadow and pasture in LUSGB. Therefore, we conducted a separate analysis, whereby the fate of all pixels of lowland meadow and permanent grassland from the historical maps (category 3 above) was determined, in terms of which broad category they are classed as in the modern land-use map (Fig. 1d).

We identified substantial changes in land-use across Great Britain during the second half of the 20th century (Fig. 1c–i). Arable land cover increased by 23% (from 22% to 27% total land cover), urban cover by 17% (from 4% to 5%), woodland cover by 91% (from 6% to 12%), and agriculturally-improved grassland increased from trace levels[16] to some 24% of the land area of Great Britain (Fig. 1c, i). These changes were largely at the expense of grassland, which has declined by 53% overall. Most significantly, semi-natural lowland meadow and pasture, included in the broader grassland category, which covered 32% of Britain according to the historical maps, shrank by 90% over the 75 year time period (Fig. 1c, h). Nonetheless, there is strong regional variation. The rate of land conversion, measured as the fraction of $25 \times 25$ m pixels within a $10 \times 10$ km grid square changing category across time periods, varied from almost complete retention of the historical land use to near complete conversion to other categories (Fig. 2b).

## Climate warming

UKCP09 monthly mean temperature grids for Great Britain were downloaded from the UK Met Office at $5 \times 5$ km spatial resolution and resampled at the $10 \times 10$ km level. Rates of temperature change (°C/decade) were then calculated by fitting a linear model in R ('stats[35]:: lm') to the time series for each $10 \times 10$ km grid square, extracting the slope value, and multiplying to ensure a denominator of 10 years. The time windows for calculation were set to the intervening period between the recording episodes for each taxon, such that warming estimates for plants were based on data for 1961−99, birds for 1973−2006 and Lepidoptera 1961−2004. Although these differ somewhat from the windows for the land-use change data- which are based on the only available data at the national level prior to large-scale grassland loss and agricultural intensification[17]- they were chosen to best represent the exposure of each taxon to climate change, and they reduced the possibility of important climate shifts between the historical and modern recording periods going uncaptured by the climate change data we used to represent them.

## Biodiversity change

A total of 20,649,112 species occurrence records at the $10 \times 10$ km grid square level (or finer) were extracted from the databases of national recording schemes for plants (data holder: Botanical Society of Britain and Ireland), birds (British Trust for Ornithology) and Lepidoptera (Butterfly Conservation) by taxon-level experts. Records collected at a resolution finer than $10 \times 10$ km were resampled to $10 \times 10$ km. We used two time periods to identify grid squares in which species had persisted in, colonised into, or been extirpated from. Bespoke time periods for each taxon were identified that: (a) matched the temporal coverage of the land-use data as closely as possible, but also (b) reflected any systematic national recording efforts for each taxon where possible (so-called 'atlas recording periods'). For plants these were 1930−60 and 2000−15 (inclusive), Lepidoptera 1930−60 and 2005−09, and birds 1968−72 and 2007−11. We only accepted records of a change in status for a particular species (i.e., an extirpation or a colonisation) in a particular $10 \times 10$ km grid square where another species of the corresponding taxon had been recorded as present during the period that the species in question was not recorded. We also limited our analyses to species with more than 100 unique $10 \times 10$ km grid square records in the first time period. The final database consisted of a total of 1192 species, including birds ($n = 137$ species), butterflies ($n = 47$ species), macromoths ($n = 333$ species) and plants ($n = 675$ species).

Despite these efforts, historical distribution data are subject to some variability in recorder effort- a form of spatial bias in the underlying species' occurrence data[43] that can influence resulting estimates of distribution change. To account for the heterogeneity in our species data in space and over time, we used the well-established FREquency SCAling LOcal program[19] (aka 'Frescalo') to estimate the recorder (sampling) effort in the neighbourhood of each grid square ($1/\alpha_i$) during the first time period, fitting this number as a taxon-specific recorder effort control in all statistical models. The Frescalo method defines a neighbourhood around each observation point ($10 \times 10$ km grid square), and weights nearby grid squares based on: (a) geographic proximity, and (b) Sorensen similarity in the biological community. Recorder effort is then calculated based on the principle that grid squares that are closer to each other are more likely to be biologically (and environmentally) similar to one another. Because the recorders of each taxon are likely to record any species of that taxon on a particular visit, the impact of chance non-detections of particular species (due to lower effort) is reduced. The Frescalo workflow was developed specifically for handling biological records data from Great Britain, and has shown to be among the best performing means of generating robust estimates of change from opportunistic species' occurrence data[44]. As such, we used its default settings wherever applicable (number of

neighbours = 100). We set target values of the standard neighbourhood frequency (Φ) for each taxon, namely: birds = 0.92, butterflies = 0.86, macromoths = 0.67, and plants = 0.75. These were values suggested by the Frescalo program[19], and were based on the properties of the data for each respective taxon.

## Identifying habitat specialisms

We consulted the species accounts within the European Red List for species[22] to identify habitat specialisms within our study taxa. Of the 466 study species with an entry on the red list (i.e., assessed species, including species of least concern LC), we determined that 176 were specialised on a single habitat (and no other), based on the habitat description given for each species in the assessment. Due to low numbers of urban and arable specialist species (n = 8 total), we limited our analysis to grassland or woodland habitat specialists (n = 28 birds, n = 118 plants, n = 22 butterflies). Moths were not assessed by the ERLs.

## Statistical modelling

To model taxon-level responses to land cover and climate change statistically (Table S2), we utilised logistic linear mixed-effects models[45] (LMMs, function 'lme4:: glmer', logit link), with the persistence (1) or extirpation (0) of populations in each 10 × 10 km grid square as the response variable and species identity as a random effect (intercept). We included controls for: (1) recorder effort (see above), (2) microclimatic variability- which has been shown to affect the long-term turnover of species in this region[18], and (3) spatial autocorrelation- by including the first two eigenvectors of a principal coordinates analysis derived from a neighbour matrix of the grid square centroids (function 'vegan[20,46]:: pcnm'). The latter inclusion accounts for the possibility that the relationships identified were an artefact of spatial autocorrelation in one (or a number) of the variables analysed[47]. To identify if temperature change, land conversion (and their interaction) were important predictors of persistence/extirpation, these variables were included or omitted in different model formulations (as per the paragraph below).

We tested five different model formulations for each taxon and selected the best performing model via AIC[19]. These formulations were: (1) "controls only" (persist/extirpate ~ recorder effort + microclimatic variability + spatial autocorrelation controls), (2) "Land conversion only" (persist/extirpate ~ land conversion + recorder effort + microclimatic variability + spatial autocorrelation), (3) "temperature change only" (persist/extirpate ~ temperature change + recorder effort + microclimatic variability + spatial autocorrelation), (4) "land conversion + temperature change additive" (persist/extirpate ~ land conversion + temperature change + recorder effort + microclimatic variability + spatial autocorrelation), and (5) "land conversion * temperature change interactive" (persist/extirpate ~ land conversion * temperature change + recorder effort + microclimatic variability + spatial autocorrelation). Species identity was included as a random effect (intercept) at all times. Confidence intervals were estimated via the 'confint' function in R ('stats[35]:: confint') with the number of simulations (bootstraps) set to 100. R-squared estimates for marginal and conditional effects were estimated via the r.squaredGLMM function in R ('MuMIn[48]:: r.squaredGLMM').

For species-level modelling (Fig. 2b, c, Fig. 3), we constructed a model set for each species equivalent to the five formulations in the above paragraph, again selecting the best performing model in each set via AIC (Fig. S1). Where cases of complete separation occurred (n = 39 species) – where persistences/extirpations formed two non-overlapping cohorts along the observed values of a predictor –the species in question was removed from our analyses, both at the species level and at taxon level. These 39 species tended to be common widespread species with very high rates of persistence, and as such were unlikely to be responsive to environmental change. These species were also excluded from the total numbers of species in each taxon

quoted in the 'Biodiversity change' section above. AME sizes for land conversion and temperature change (Fig. 2b, c) were estimated via 'margins[49]:: margins'. Species with only one environmental change variable (land conversion or temperature change) in their 'best' model, or both environmental change variables in an additive (non-interactive) model contributed values to Figs. 2b, c and 3 (n = 668 total).

For the 230 species for which the model with an interaction term performed best, we first used this full interactive model to predict persistence across the range of observed changes in temperature change and land conversion in Great Britain, including the minimum and maximum value for each predictor, and each decile (10th, 20th, ..., 90th percentile), generating 11 × 11 = 121 estimated values for each species. All control variables were set to their median amount. Second, we generated a further, analogous set of 121 estimated values at the same levels of temperature change and land conversion using an otherwise equivalent model without an interaction term (an additive model). Third, to ascertain if the two series of predictions for each of the 230 species were different, Paired t tests were performed, and a Bonferroni-corrected p-value threshold of 0.000217 was applied (0.05/230). Differences in persistence probability estimates across the two models were further investigated by sorting the differences at each decile (rows in Table S4) into size categories of difference in estimated persistence (±%, columns) and calculating how many species fell within each category- thus each row adds up to 230 species.

We checked for collinearity between our explanatory variables by conducting a number of Pearson correlation tests at the grid square level. First, we checked for correlations between the two environmental change variables- land conversion and climate warming. Correlations were significant (at p < 0.05), but at values of r (<0.7) which will not result in spurious interpretations[50]: Plants (1961–1999), r = 0.58, d.f. = 2654, p < 0.0001; Birds (1973–2006), r = 0.38, d.f. = 2654, p < 0.0001; Lepidoptera (1961–2004), r = 0.50, d.f. = 2654, p < 0.0001. To test if squares subject to higher rates of land conversion were also subject to higher recorder effort (them being more proximate to roads, cities and/or human population), we then correlated land conversion with our taxon-specific recorder effort measures. Here again we found significant (p < 0.05) correlations but at weaker values of r (all values of r < 0.5): Plants, r = -0.04, d.f. = 2705, p = 0.01; Birds, r = 0.46, d.f. = 2785, p < 0.0001; Butterflies, r = 0.33, d.f. = 1673, p < 0.0001; Moths, r = 0.19, d.f. = 1483, p < 0.0001. Finally, because baseline climate can also influence persistence probability (vis-à-vis change in the climate), we also tested for a correlation between mean annual temperature and temperature change (for 1961–2006, the maximal window over which we calculated change across the taxa). Here again we found a significant but weak correlation (r = 0.56, d.f. = 2654, p < 0.0001).

## Changes in species' range extents

To analyse each species' geographical distribution change, we again used the Frescalo approach, estimating the change in each species' geographic distribution per decade. This time, we used the implementation of August et al. 2018 ('sparta[51]:: frescalo') modified to allow the use of Sorensen similarity in neighbourhood determination using vegan[44]:: vegdist, including both the historical and modern species observations (i.e., persistences, extirpations and colonisations) to approximate historical and modern distributions. As a test of the reliability of the Frescalo approach, we also calculated a more conservative, 'Telfer' metric of distribution change, that assigns each species a value of change relative to other species[52], using 'sparta[51]:: telfer' separately for each taxon. Both metrics were in high agreement (Fig. S2) and we therefore consider our Frescalo outputs to be reliable. A Chi-square test (stats[35]::chisq.test) was used to compare whether species persistence response types were associated to positive or negative Frescalo distribution trends. We analysed 668 (non-interactive) species divided into eight groups depending on their direction

of response and whether the best-performing model was contained temperature change, land conversion or both (red, blue and purple sections of Fig. 3, Table S7). Two plant species were excluded for analysis because they exhibited distribution trends of zero: *Neottia cordata* (land conversion only, negative) and *Eleocharis quinqueflora* (additive model, land conversion and temperature change both negative). A full list of species analysed and their persistence rates is provided in Supplementary Data 1.

### Reporting summary
Further information on research design is available in the Nature Portfolio Reporting Summary linked to this article.

## Data availability
The land-use data generated and analysed in this study have been deposited online at https://doi.org/10.5878/9wks-qg91. The full resolution land-use data remain under copyright and are not available due to data privacy laws. The climate data used in this study are available on the UKCIP18 database at https://ukclimateprojections-ui.metoffice.gov.uk. The biodiversity data used in this study are available on the NBN Gateway at https://nbnatlas.org.

## Code availability
Custom scripts for generating results are available online at dx. https://doi.org/10.6084/m9.figshare.23925462.

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

## Acknowledgements

We thank the many thousands of volunteer surveyors of land use and biodiversity, without whom these analyses would not have been possible. Data and imagery from the Land-Utilisation Survey of Great Britain are reproduced with permission of the copyright holder Giles N. Clark. Chris Fleet at National Library of Scotland provided assistance with land-use maps covering Scotland, and Natural England provided assistance with maps covering England and Wales. Thanks to Pieter De Frenne, Bronwen Whitney, Tim Newbold, Tomas Pärt and Mike Rogerson for providing comments on the manuscript. This work was supported by a UKRI Natural Environment Research Council grant (NE/M013030/1, for A.J.S., J.K.H. and C.D.T.), a Northumbria University Vice Chancellor's Senior Research Fellowship (for A.J.S.), and grants from the Swedish Research Councils Formas and VR (2015-1065 and 2020-04276, for A.G.A.).

## Author contributions

Conceptualisation: A.J.S., A.G.A., C.D.T., J.K.H. Methodology: A.J.S., C.J.W., P.A., H.S., A.G.A., N.J.B.I. Data provision: R.F., N.J.B.I., K.J.W., B.M. Visualisation: A.J.S., A.G.A. Funding acquisition: A.J.S., J.K.H., H.S., C.D.T., A.G.A. Expert guidance: A.J.S, C.M.B., J.K.H., N.J.B.I., C.D.T., K.J.W., A.G.A. Writing – original draught: A.J.S., A.G.A. Writing – review & editing: A.J.S., C.J.W., P.A., C.M.B., R.F., J.K.H., N.J.B.I., B.M., H.S., C.D.T., K.J.W., A.G.A.

## Competing interests

The authors declare no competing interests.
