## [Peer Review File · Nature Communications]

Linking climate warming and land conversion to species' range changes across Great BritainREVIEWER COMMENTS

Reviewer #1 (Remarks to the Author):

This manuscript presents noteworthy results in modelling the effects of temperature changes, land conversion, and their interactions on persistence versus extirpation for four major taxonomic groups (birds, butterflies, macromoths, and plants) in Great Britain over a period of ~75 yrs. The work compares to previously established literature in that distributional changes have been documented relative to climatic changes, but it adds the effects of land conversion and the interactive effects of land conversion and climate change (temperature in this case). The authors found that individual or additive effects of climatic warming and land conversion increased the probability of persistence for 40% of the species, but decreased it for only 12%. Notably, interactive effects were rare.

The work supports the conclusions and claims, the methodology is sound, and there is generally enough detail provided for results to be reproduced. The data sets used in these analyses are robust (20 million biological records covering historical 1930-1972 and modern 2000-2015 time periods at a grid scale of 10 x10 km or finer across Great Britain). The authors tested carefully for recorder effort influences using an established Frescalo method. Climatic data of mean monthly temperatures were resampled to 10 x 10 km grids. The authors took care to avoid having rare species skew the results, excluding species recorded in 100 unique 10 x 10 km grid squares or fewer during the first period. There was also a separate analysis of grassland landcover change to carefully avoid the potential for obscuring results due to differences in historic vs. modern grassland landcover data (Lines 242-244).

Their approach of using historical maps and checking for differences in results based upon different methods and different mapping approaches is reasonable. The 25 x 25 m pixel land-use change analysis within each 10 x 10 km grid cell provides important finer scale insight on geographical locations of land conversion. They wisely used a Bonferroni correction when assessing effects at the species level, given the large number of individual tests performed. Similarly, they checked for collinearity between explanatory variables by conducting Pearson correlation tests with a cut-off of $r > 0.7$.

This paper is in good order in terms of reporting factual results. However, I have several areas where I would suggest improvements, and have listed them by manuscript section below.

Abstract: The abstract is written in very broad brushstrokes, but is not as easy to interpret with respect to several important details. For example, what are the implications that warming and land conversion increased the probability of persistence of species? And did range shifts move in a particular direction? Were results similar across taxa?

Methods: One question I have regarding the methods has to do with the justification for the time windows used for calculating climate warming. In lines 271-273, the authors state the "The time windows for calculation were set to the intervening period between recording episodes for each taxon." This means that the climatic changes were assessed for a subset of the 75 years (plants 1961-99, birds 1973-2006, and Lepidoptera 1961-2004). While this could be deemed a conservative approach to assessing temperature changes over time (38, 33, and 43 yrs. respectively), it is not the full 75 years used for assessing landcover change. This means that the temperature and the landcover data for historic and modern time periods are not exactly aligned. The authors can probably justify this decision, but that justification should be provided.

Discussion: My major comments with regard to the discussion have to do with suggesting that the authors add details to explain the implications of these results to ecologists and land managers. For example, why might the combination of temperature change and land conversion create more favorable conditions for species persistence? This result was found as additive effects for butterflies, macromoths, and plants (line 95), and interactive effects for birds. If GB is generally a cool, temperature-limited system, one might imagine that warming could create more favorable conditions for productivity and survival in some of these species. The authors could comment on whether that is applicable (and I do note that some of the species distribution changes were associated with negative temperature changes). However, even more importantly, the fact that land conversion was associated with higher species persistence calls for more discussion. These results would seem contrary to expectations, particularly from the grassland perspective. The

authors found that agriculturally improved grasslands increased from trace levels to 24% of the land area, woodland cover increased from 6 to 12%, and semi-natural grassland decreased from 65-30%. Given the type of changes measured here, how might this affect suitable habitat for the taxa examined? With substantial loss of semi-natural grassland, I would have expected a stronger negative signal in grassland obligate species. The authors mention that the results described here do not reflect historical extirpations resulting from land use change that occurred prior to the period of their study (Line 170-172). Could it be that many of the grassland obligates are already extirpated? Alternatively, given that semi-natural grassland primarily transitioned to "improved grassland" or "arable" landcover, what might this imply with respect to creation of potential new habitat for other more generalist species? I realize that the data sets did not include generalist vs. specialist categorizations for the species examined, but some comments based on knowledge of the species-habitat associations would make these summaries more valuable. Relatedly, for responses to land conversion, there was a large cohort of weakly positive responses and smaller cohort of strongly negative responders (Line 115-117). Can the authors make any generalizations about these groups, particularly those with strong negative responses? Finally, it is interesting that meaningful interactions between land use change and temperature change on species persistence were relatively rare (Line 137). The authors state that "land use change can also affect the capacity of species to respond to climate change" (lines 46-47). And yet, the results did not support this expectation. Can the authors discuss the implications of this finding?

Summary: With respect to the final comments in the manuscript, I would not include the topic of microclimatic refugia in this paper because there is not enough detail in the assessment of the species and taxonomic-level responses to address microclimatic refugia and this topic is not addressed earlier in the manuscript. I would also encourage the authors to add comments about the implications of these results as described above in discussion section.

Specific Editorial Comments:

Line 64: Define "agriculturally-improved grassland." Is this a grassland into which seeds of agriculturally cultivated grass species are planted to improve forage for grazers?

Lines 170-173: The statement that "Management changes within land use categories, such as agricultural intensification, may have also been detrimental..." is possibly true, but there are no data to substantiate.

Line 346: change "predicted" to "predict"

Figure 3 lumps all species across taxa in summarizing responses to land conversion, temperature change, and the combination thereof. Although this is a nice synthesis, it calls out for a supplementary table that separates out responses by taxonomic group.

Diane Debinski

Reviewer #2 (Remarks to the Author):

This manuscript estimates the effects of climate warming and land conversion on over 1000 species in Great Britain between two time periods (1930-1972) and (2000-2015). Overall, this is an impressive data set and the questions being addressed are important for conservation. I have the following general comments.

First, land-use change is a complex phenomenon that can alter biodiversity in a wide variety of ways. Here, for testing biodiversity responses to land-use change, the authors simply summarized change over time, not distinguishing the types of change arising (despite showing land-use specific changes in general in Fig. 1e). This general metric does not provide much insight into the problem and might even muddle interpretation. For example, it appears most of the land-use change was driven by grassland loss, but in pooling information the authors do not distinguish this type of change from other types of change on biodiversity nor the expected outcomes. As grassland communities are often not as species rich as forested communities, the high frequency of positive responses to 'land-use change' in general are not surprising but without dissecting these issues, the patterns shown are hard to interpret. A related point is that the current predictor is simply the fraction of land-use change; however, this ignores current land-use. For example, 10 x 10 km grids could suffer the same fraction of change, but depending on baseline conditions of land-use, the consequences of such changes are likely fundamentally different as it is well known that land

change effects can have strong threshold effects on species. This manuscript would be much more insightful if the authors more explicitly probed the types of land change and linked them to knowledge of species habitat attributes (e.g., did grassland species decline?).

Second, the authors argue that despite model selection providing evidence for interactions for a substantial portion of species, these effects are not meaningful based on similarity of predictions. This issue could be better understood and articulated by more clearly distinguishing between research goals: inference versus prediction. In addition, for prediction, the case would be much more convincing if the authors actually used some sort of predictive assessment, such as k-fold cross validation rather than a paired t-test. Currently, simply comparing predictions doesn't help the reader understand the extent to which predictions are reliable or vary in space/time. There is a large body of literature on assessing predictive capacity of models (e.g., in the forecasting literature, species distribution modeling literature) that this manuscript would benefit from including.

Third, while this data set is quite impressive, there are fundamental challenging in estimating changes over time with data like these. The authors rely entirely on using the Frescalo method for addressing this issue, but it is not clear to me that it is sufficient. As I understand it, this method attempts to account for effort but it does not provide any information on spatial sample bias, which is essential for interpreting land-use change effects (that inherently vary in space). The authors need to better justify that this method is sufficient for both effort and spatial sample bias. In addition, it would be helpful to discuss any issues that arise when comparing across these time periods that also vary considerably in the time frames. I think the latter is probably less important for within taxon comparisons, but it does influence the general interpretation of the estimates and, given that sampling time frames vary across taxa, it makes direct comparisons across taxa challenging.

Reviewer #3 (Remarks to the Author):

An interesting paper using an exciting new dataset on land use change (which does look amazing) to look at how climate change and land use change might impact persistence in 4 groups. So while I like this idea, I found the paper quite hard work and have lots of questions and queries, some pretty fundamental to the conclusions you draw. Apologies if I have the wrong end of the stick in places.

Line 29. It seems odd to say that evidence for additive or interactive effects is lacking – quite a few recent papers have explored and shown these effects – in vertebrates and insects etc...

Line 33 You say, 'to distribution changes' but you only model persistence.

Line 37. When you say, 'interactive effects were rare, being detected in less than 1 in 5 species.' An occurrence of 20% sounds high to me and of biological relevance. Rephrase/explain.

Line 37. When you say, 'Such individualistic responses emphasise the importance of including species-level information in policies targeting biodiversity and climate adaptation.' Spell out how you suggest that is done. It reads as a throw-away comment.

Line 52. Agree that efforts have been limited, there are big data problems, but why not reference here the relevant work that has been published?

Line 53. The new land use change map looks extremely useful and is carefully constructed. Amazing. I love the new maps on their own.

Line 73. I like figure 1 but would like 1F to show the separate mapped changes in each of the 5 different land use on GB maps. Their pattern of spatial change would be extremely useful to visual and understand.

Line 79. Here you say you excluded species with 100 or fewer records but for which period? Later you say the first period, which I query, because that excludes any natural colonists? Please clarify.

Line 83. If this paper only relates to persistence, then change the title.

Line 84. What about the opposite case of absence in the first period but presence in the second? Is that considered?

Line 97. Interesting to see extirpations were lower where land conversion was also high, would you not predict the reverse? There's lot of evidence to show that the land use changes and the conversions you describe were associated with specific and general abundance and distributional declines. Are you testing particular hypotheses here and making predictions of effects? That might

help structure the paper and help the reader?

Line 102. The individual species models now seem to duplicate the results presented in the previous paragraph in a slightly more jumbled fashion. I see these feed into fig 2b and 2b. Is this better presented as supplementary analyses?

Line 102. The other piece of basic information relevant here, but missing from what I can see, would be a tabulation of the presence and absence by each taxonomic group. I am interested to understand the patterns of distributional changes in these taxa, that then sit behind the correlations. These changes are hinted at in the paper but unclear.

Line 114. Is there a word missing here?

Line 152. You say, 'Taken in sum, our results are compatible with the highly variable, yet overall neutral trends in biodiversity at the local and landscape scales across the globe' but how so? Expand and explain. You aren't measuring biodiversity change per se, just range not abundance, lots of caveats in using unstructured presence data, attempting to correct for sampling effort, just four taxa etc. I suggest you delete this sentence and such speculative comparisons. It adds confusion.

Line 168. Is it safe to say, 'our results suggest that climate change has been a more important driver of population persistence over this period than land-use change', first given the data quality and just four taxa groups, plus the methods used, second, the inherent difficulty in quantifying the two in an equivalent way in relation to species responses? They are apples and oranges. A small change in land use, imperceptible when considering such broad categories of land use, change might determine the fate of a species. The extent of a land type may have no bearing on the intensity of land use within in - and the latter may be critical to species. I suggest you are more circumspect here and cautious and include some discussion of the real caveats here.

Line 173. Rather underwhelming and quite misleading to say, 'Management changes within land use categories, such as agricultural intensification, may also have been detrimental in recent decades.' with no reference to a body of research that shows such effects in a range of different ways. The emphasis on 'may' here but the whole paragraph feels unbalanced.

Line 177. Add references to the work showing those effects.

Line 186. The final sentence here, 'These factors,' is again a somewhat through-away comment. Spell out what is intended in plain language and be more specific in your recommendation.

Line 287. I can see why you limited your analyses to species with more than 100 unique 10 × 10 km grid square records in the first time period, but why not also allow the same for the second period? Your methods seem to exclude colonising species, which seem highly relevant.

Line 307. Why model taxon-level responses to land cover and climate change statistically? That is, why fit separate models for each taxa? Why not one model with taxa as a random or fixed effect? The thing you are interested in is the effects of climate warming and land conversion on species' range change – not testing differences between taxonomic groups etc.

Line 307. The models need to incorporate correction for phylogenetic similarity or at least test that such effects are not biasing your results.

Line 333. Why now embark on individual overlapping species-level modelling of the same effects?

Line 336. Explain in more specific detail how perfect prediction occurs in this case and how it affects your analysis of biodiversity change.

Line 365. You report that higher rates of land conversion may also have been subject to higher recorder effort – that would explain artefactually why you find, rather surprisingly, positive relationships between persistence and land conversion (Fig 2). That is a major worry.

Line 370. How does the correlation between mean annual temperature and temperature change influence your findings? You describe this as a weak correlation ($r = 0.56$, $p < 0.0001$) - but many would kill for such a correlation.

Line 575. Fig 2 – no correction for phylogenetic effects here.

Line 584. Fig 3 – again no correction for phylogenetic effects here, which is problematic as the counts of numbers of species showing effects is being presented, regardless of their relatedness.

RD Gregory

Response from the author team: Thanks to the editing team and reviewers for considering our manuscript, and for the chance to improve it by acting on the thoughtful comments set out in the reviews. We provide a point-by-point response below, and have used track changes on the MS to make it clear where we have made edits in response to suggestions. We hope that you find the manuscript much improved as a result of these changes, and thanks again for this opportunity.

Reviewer #1 (Remarks to the Author):

This manuscript presents noteworthy results in modelling the effects of temperature changes, land conversion, and their interactions on persistence versus extirpation for four major taxonomic groups (birds, butterflies, macromoths, and plants) in Great Britain over a period of ~75 yrs. The work compares to previously established literature in that distributional changes have been documented relative to climatic changes, but it adds the effects of land conversion and the interactive effects of land conversion and climate change (temperature in this case). The authors found that individual or additive effects of climatic warming and land conversion increased the probability of persistence for 40% of the species, but decreased it for only 12%. Notably, interactive effects were rare.

The work supports the conclusions and claims, the methodology is sound, and there is generally enough detail provided for results to be reproduced. The data sets used in these analyses are robust (20 million biological records covering historical 1930-1972 and modern 2000-2015 time periods at a grid scale of 10 x10 km or finer across Great Britain). The authors tested carefully for recorder effort influences using an established Frescalo method. Climatic data of mean monthly temperatures were resampled to 10 x 10 km grids. The authors took care to avoid having rare species skew the results, excluding species recorded in 100 unique 10 x 10 km grid squares or fewer during the first period. There was also a separate analysis of grassland landcover change to carefully avoid the potential for obscuring results due to differences in historic vs. modern grassland landcover data (Lines 242-244).

Their approach of using historical maps and checking for differences in results based upon different methods and different mapping approaches is reasonable. The 25 x 25 m pixel land-use change analysis within each 10 x 10 km grid cell provides important finer scale insight on geographical locations of land conversion. They wisely used a Bonferroni correction when assessing effects at the species level, given the large number of individual tests performed. Similarly, they checked for collinearity between explanatory variables by conducting Pearson correlation tests with a cut-off of $r > 0.7$.

This paper is in good order in terms of reporting factual results. However, I have several areas where I would suggest improvements, and have listed them by manuscript section below.

Response from the author team: Many thanks for considering our manuscript so carefully. We have adopted a number of your suggested improvements and describe how we did this below.

Abstract: The abstract is written in very broad brushstrokes, but is not as easy to interpret with respect to several important details. For example, what are the implications that warming and land conversion increased the probability of persistence of species? And did range shifts move in a particular direction? Were results similar across taxa?

Response from the author team: We agree that the abstract is broad, and although we were striving for a balance between clarity of language and faithfully representing the individualistic nature of the 1,192 species we studied, we have revised it as per the requested details. Specifically on your examples, we have now made it clear that the effect of the environmental change variables was obvious in range shifts of both directions (an important point), but we consider it beyond a 150 word abstract to explain what the implications are of the overall increase in persistence. Yet, in response to this and your further point below we have expanded on this in the discussion (alongside the addition of a number of other provisos).

Re: result similarity across the taxa. In response to Reviewer #3's suggestions re- possible confusion between taxon- and species-level analyses, we have now deprecated the taxon-level results, and moved them to the Supplementary Materials. (Incidentally, results largely were similar across taxa when looking at the outputs of the models fitted at the taxon level or the species level. Whilst we do think this is a strength of the MS, with the abstract limited to 150 words, we don't feel that we can fit it in without removing other material that is more important to highlight from the MS).

Methods: One question I have regarding the methods has to do with the justification for the time windows used for calculating climate warming. In lines 271-273, the authors state the "The time windows for calculation were set to the intervening period between recording episodes for each taxon." This means that the climatic changes were assessed for a subset of the 75 years (plants 1961-99, birds 1973-2006, and Lepidoptera 1961-2004). While this could be deemed a conservative approach to assessing temperature changes over time (38, 33, and 43 yrs. respectively), it is not the full 75 years used for assessing landcover change. This means that the temperature and the landcover data for historic and modern time periods are not exactly aligned. The authors can probably justify this decision, but that justification should be provided.

Response from the author team: Thanks for this considered point, and as the reviewer alludes to, comparisons across the taxa required a conservative approach to including a test for climate change that both allows for consistency, yet is also sensitive to potentially important climate shifts between biodiversity recording periods that could go uncaptured should we opt for another approach (e.g. a window of 1972-2000 for all taxa). New wording reads: 'Although different from the land-use data windows, which are based on the only available data at the national level prior to large-scale grassland loss and agricultural intensification¹⁵, these windows were chosen to best represent the exposure of each taxon to climate change, and they reducing the possibility of important climate shifts between the historical and modern recording periods going uncaptured by the climate change data we used to represent them'.

Discussion: My major comments with regard to the discussion have to do with suggesting that the authors add details to explain the implications of these results to ecologists and land managers. For example, why might the combination of temperature change and land conversion create more favorable conditions for species persistence? This result was found as additive effects for butterflies, macromoths, and plants (line 95), and interactive effects for birds. If GB is generally a cool, temperature-limited system, one might imagine that warming could create more favorable conditions for productivity and survival in some of these species. The authors could comment on whether that is applicable (and I do note that some of the species distribution changes were associated with negative temperature changes).

Response from the author team: Many thanks for this comment on the discussion (and the further points that follow). We would highlight that the discussion you reviewed at this first submission stage was somewhat short because it was tailored to another Nature Family journal with stricter constraints on space. Now, and in response to this comment (and the other discussion points below), we make use of the additional space afforded to articles in Nature Communications to expand on the detail we include here (continuing to explain specifically how we do this on a point-by-point basis).

In response to this specific point, although we have now deprecated the taxon-level results referred to by Reviewer #1 here, the pattern of an overall (mild) positive effect of environmental change is also present at the species level. We have therefore added a few sentences to the general discussion to more completely address this, which read as follows: 'The potential discrepancy between our findings of overall weakly positive responses to environmental change and those of others^{2,8} might also relate to our focus on range retractions, as opposed to overall changes in geographic range and/or abundance. This is because the processes that control the survival of populations and those that control other population characteristics (dispersal, density, competition, etc) may differ. For example, a relatively cool-adapted plant species reaching its warm (equatorward) range limit in our study area may be simultaneously: a) declining in abundance throughout most of its British range; b) be able to (at least temporarily) persist in the majority of its historical range; and c) be temporarily alleviated from interspecific competition from warm-adapted species that have thusfar been unable to colonise (due to high rates of isotherm shifts), as well as the improved conditions for growth and/or survival that warmer temperatures would bring'.

However, even more importantly, the fact that land conversion was associated with higher species persistence calls for more discussion. These results would seem contrary to expectations, particularly from the grassland perspective. The authors found that agriculturally improved grasslands increased from trace levels to 24% of the land area, woodland cover increased from 6 to 12%, and semi-natural grassland decreased from 65-30%. Given the type of changes measured here, how might this affect suitable habitat for the taxa examined? With substantial loss of semi-natural grassland, I would have expected a stronger negative signal in grassland obligate species. The authors mention that the results described here do not reflect historical extirpations resulting from land use change that occurred prior to the period of their study (Line 170-172). Could it be that many of the grassland obligates are already extirpated? Alternatively, given that semi-natural grassland primarily transitioned to “improved grassland” or “arable” landcover, what might this imply with respect to creation of potential new habitat for other more generalist species? I realize that the data sets did not include generalist vs. specialist categorizations for the species examined, but some comments based on knowledge of the species-habitat associations would make these summaries more valuable.

Relatedly, for responses to land conversion, there was a large cohort of weakly positive responses and smaller cohort of strongly negative responders (Line 115-117). Can the authors make any generalizations about these groups, particularly those with strong negative responses?

Response from the author team: We thank the reviewer for these careful points, and in response to the ‘relatedly’ point (and also to Reviewer #2’s related point re- specialists), we have added a new set of analyses and text to the results section, specifically tackling the question of if species with habitat specialisms were more responsive to changes in their habitat. Indeed, we found that (for example) grassland specialist plants responded positively to retention of grassland habitat (which would generally contribute to lower levels of habitat conversion).

We have also expanded on habitat specialisms in the discussion, and now include some possible mechanisms for the effects we detected (alongside some careful caveats). Revised text reads: ‘We showed that habitat losses have had a negative effect on species’ persistence for habitat specialists (Fig. S3, S4), but in the case of our broader species pool (including generalists), species varied widely in their response, and the balance of responses to land conversion was weakly positive. This may reflect: a) the relative lack of habitat specialists amongst our study species (or study region) and/or the extirpation of specialist species prior to our study period, b) biotic mechanisms that support population persistence (if not abundance) in environments that are disturbed²⁴, heterogeneous²⁵, or in dynamic equilibrium²⁶. In either case, we would emphasise that environmental change is unlikely to be beneficial to species’ persistence per se, and indeed, space-for-time approaches suggest that interactive effects of climate and intensive agriculture may drive widespread declines in insect biodiversity⁸, particularly in the tropics’.

Finally, it is interesting that meaningful interactions between land use change and temperature change on species persistence were relatively rare (Line 137). The authors state that “land use change can also affect the capacity of species to respond to climate change” (lines 46-47). And yet, the results did not support this expectation. Can the authors discuss the implications of this finding?

Response from the author team: As per this request we have expanded our discussion of this finding: ‘Interestingly, although land conversion and temperature change often acted on species persistence in the same direction (i.e. ++ or --; Fig. S1), meaningful interactive effects were found to be relatively rare (Fig. S1, Table S6). This perhaps surprising result is not unprecedented in multispecies studies (e.g. in pollinators²³), and implies that the mechanisms by which the changed conditions affect species, populations, and individuals are different, or at least, are not susceptible to compounding or accelerating effects of multiple drivers acting at once’.

Summary: With respect to the final comments in the manuscript, I would not include the topic of microclimatic refugia in this paper because there is not enough detail in the assessment of the species and taxonomic-level responses to address microclimatic refugia and this topic is not addressed earlier in the manuscript.

Response from the author team: Thanks- we have deleted the point on microclimatic refugia.

I would also encourage the authors to add comments about the implications of these results as described above in discussion section.

Response from the author team: Thanks- we hope that our response to the first discussion point satisfies this request.

Specific Editorial Comments:

Line 64: Define “agriculturally-improved grassland.” Is this a grassland into which seeds of agriculturally cultivated grass species are planted to improve forage for grazers?

Response from the author team: We did include a definition in the methods, but we should have made this more explicit. Revised text now reads: ‘and agriculturally-improved grassland (here defined as intensively-managed and fertilised pastures that are highly productive but low in biodiversity)’.

Lines 170-173: The statement that “Management changes within land use categories, such as agricultural intensification, may have also been detrimental...” is possibly true, but there are no data to substantiate.

Response from the author team: Agreed- and also in response to Reviewer #3’s comment, we have now caveated this statement more carefully. Text now reads: ‘Management changes within land use categories (as such beyond our scope here), such as agricultural intensification, have also been shown to be detrimental in recent decades (e.g. for birds²⁸), and could surpass land conversion as stronger drivers of change in regions (such as Great Britain) where land-use configurations are now largely stable’.

Line 346: change “predicted” to “predict”.

Response from the author team: Changed as requested, thanks.

Figure 3 lumps all species across taxa in summarizing responses to land conversion, temperature change, and the combination thereof. Although this is a nice synthesis, it calls out for a supplementary table that separates out responses by taxonomic group.

Response from the author team: Thanks for this suggestion- we have now added a supplementary table with the numbers of species according to response category and sign of distribution change (Table S7). Note that we do not apply any statistical analyses separately by taxonomic group, due to the low numbers of species within many groups precluding any meaningful analysis of observed vs. expected groupings, e.g. 31 Butterfly species split across the eight response types and two distribution directions.

Diane Debinski

Reviewer #2 (Remarks to the Author):

This manuscript estimates the effects of climate warming and land conversion on over 1000 species in Great Britain between two time periods (1930-1972) and (2000-2015). Overall, this is an impressive data set and the questions being addressed are important for conservation. I have the following general comments.

Response from the author team: Many thanks for your kind words on our dataset and the questions we apply it to.

First, land-use change is a complex phenomenon that can alter biodiversity in a wide variety of ways. Here, for testing biodiversity responses to land-use change, the authors simply summarized change over time, not distinguishing the types of change arising (despite showing land-use specific changes in general in Fig. 1e). This general metric does not provide much insight into the problem and might even muddle interpretation. For example, it appears most of the land-use change was driven by grassland

loss, but in pooling information the authors do not distinguish this type of change from other types of change on biodiversity nor the expected outcomes. As grassland communities are often not as species rich as forested communities, the high frequency of positive responses to 'land-use change' in general are not surprising but without dissecting these issues, the patterns shown are hard to interpret.

A related point is that the current predictor is simply the fraction of land-use change; however, this ignores current land-use. For example, 10 x 10 km grids could suffer the same fraction of change, but depending on baseline conditions of land-use, the consequences of such changes are likely fundamentally different as it is well known that land change effects can have strong threshold effects on species. This manuscript would be much more insightful if the authors more explicitly probed the types of land change and linked them to knowledge of species habitat attributes (e.g., did grassland species decline?).

Response from the author team: We agree that land-use change is complex in itself, and its effects- when it comes to the analysis of its effects on >1,000 species even more so- and so we would argue that to fully determine these effects for even a small proportion of species is beyond the scope of a single manuscript. This is without considering that there are also multiple different climate change variables that species will be responding to, and their varying temporal elements (e.g. van de Pol 2016 MEE). Our land conversion metric- analogous to the climate change metric- quantifies the overall 'disruptive' effect of land-use change in a landscape, and as such, we consider it both a useful overall measure of pressure from this form of environmental change, and a useful metric where a priori knowledge of species' habitat associations may not be complete (or indeed where generalist species are the subject of analyses).

Nevertheless, where habitat specialisations can be identified, we agree that the usage of a more tailored metric is likely to be more appropriate. In response to the reviewer's suggestions here, we have therefore revised and expanded our analysis of species-level responses by relating changes in habitat specialists to changes in their specific habitat. Using the species-level accounts contained within the latest versions of the European Red Lists for species (EEA 2018)- which covers birds, plants and butterflies (no Red List is available for moths, which represent 333 of our study species)- we identified species that were specialised to (only) open habitats (n = 116 open habitat specialists) and (only) woodland habitats (n = 52 woodland habitat specialists) and performed alternative, habitat-specific versions of the analyses. For these analyses, we replaced the land conversion metric with measures of change in the respective specialist habitats. We found that the retention of semi-natural grasslands (categories of which are prioritised for biodiversity conservation as part of the Habitats Directive/Regulations) is positively associated with the persistence of grassland specialists, while the increase in forest cover due to landscape abandonment and forest plantations had a weaker (but still positive) effect on woodland specialists. We include the results of these analyses in a bespoke results paragraph in the manuscript. We really think that these new analyses substantially enhance the comprehensiveness of our study, and we thank the reviewer for their thoughtful suggestion.

New paragraph now reads: 'To determine the extent to which habitat specialists were more sensitive to changes in land use (or land use-climate interactions) than the wider species pool, we conducted an additional analysis on a subset of species (n = 168 species) we identified as specialised to a single habitat within the European Red Lists for species²⁰. We re-ran the statistical models of persistence vs. extirpation formulated as before, but this time substituting the generic 'land conversion' variable for relative change in the specific habitat type each specialist was associated with. Responses of specialists were broadly similar to the full species pool (Fig. S3 vs. Fig. S1); however, for those species with habitat change in best performing models (without interactions), the AME of habitat change was larger (median 1.8% vs. 1.4% per 10% increase in cover of specialist habitat type, Fig S4). Grassland specialists appeared more sensitive, with a 10% change (more a case of retention rather than increase of semi-natural grassland habitat; Fig. 1g) resulting in a median 1.95% increase in the persistence probability. Increases in woodland cover also had a positive, albeit less strong, effect on persistence probability in woodland specialists (AME median +0.79% per 10% increase in cover). For the 30 species where an interaction effect between habitat change and climate change was included in the best model, predictions derived from that model differed from the analogous additive model (omitting an interaction term) for 5 species (Paired t-tests, d.f. = 118, Bonferroni-corrected p-value = 0.0016), again a similar proportion to results from the full species pool'.

Second, the authors argue that despite model selection providing evidence for interactions for a substantial portion of species, these effects are not meaningful based on similarity of predictions. This issue could be better understood and articulated by more clearly distinguishing between research goals: inference versus prediction. In addition, for prediction, the case would be much more convincing if the authors actually used some sort of predictive assessment, such as k-fold cross validation rather than a paired t-test. Currently, simply comparing predictions doesn't help the reader understand the extent to which predictions are reliable or vary in space/time. There is a large body of literature on assessing predictive capacity of models (e.g., in the forecasting literature, species distribution modeling literature) that this manuscript would benefit from including.

Response from the author team: Whilst the reviewer's suggestions on how to assess the reliability of model predictions, and assess their variability across space or time, are certainly the way to go in answering such questions, we would emphasise at this point that we are not trying to forecast or predict species' responses here- in the sense that a species' distribution model would do. Rather, we are interested in answering a relatively simple question: how similar are the (statistical) predictions of two different model formulations across the observed parameter space of the predictors? We therefore feel that a simple tool to compare two (paired) series of numbers- a t-test- is an intuitive and appropriate choice for answering it.

In response to this point, we have also re-written the legend and headers for Table S6 so it is clearer what was done to explore if/how the statistical predictions of the models with and without interaction terms differed (in particular parts of predicted space, if not overall, which is what the t-tests do). We hope this makes the evidence that supports our point re-interactions more intuitive and easier to follow. We would also highlight that even a focus on habitat specialists- see new results paragraph and answer to the above point- does not reveal that those species one would expect to be more sensitive to habitat change are also more prone to habitat-climate interactions- the proportions of species with interaction effects in their 'best' model, and the proportion with significantly different predictions arising from the inclusion of an interaction effect (vs analogous additive model) are the same.

Third, while this data set is quite impressive, there are fundamental challenging in estimating changes over time with data like these. The authors rely entirely on using the Frescalo method for addressing this issue, but it is not clear to me that it is sufficient. As I understand it, this method attempts to account for effort but it does not provide any information on spatial sample bias, which is essential for interpreting land-use change effects (that inherently vary in space). The authors need to better justify that this method is sufficient for both effort and spatial sample bias.

Response from the author team: We agree that studying biodiversity change over time using unstructured data is very challenging. We chose to use the Frescalo method because it was developed for biological datasets in Great Britain, has been recommended for use in data (like ours) that are split into distinct time periods (Isaac et al. 2014, Methods in Ecology & Evolution), and is regularly used to control for sampling effort in other studies using similar types of data (e.g. Auffret & Svenning 2022, Nature Communications; Eichenberg et al. 2021, Global Change Biology; Redhead et al. 2021, Ecology Letters).

In our initial submission, we focussed on describing how our methods account for change in recording effort over time, but Frescalo is also designed to correct for spatial bias, through estimating local relative frequencies of occurrence within geographically proximate grid squares, weighted according to proximity and compositional similarity. Indeed, when describing the new method, Hill (2012; Methods in Ecology & Evolution) writes "The big advantage of the neighbourhood-frequency method in estimating trends is that it corrects for uneven spatial recording over time". (i.e. differences in space and over time). We have now modified the relevant paragraph in the Methods to make sure that it is clear to the reader that Frescalo does incorporate spatial aspects in its calculation of recorder effort.

In addition to using Frescalo to estimate recorder effort for each grid square, we also include other measures in our analysis to ensure robustness of our results. First, we only analysed persistence or extirpation of species that were recorded in a grid square during the historical period where sampling effort was lower. Because in the modern period recording effort in Britain is higher (e.g. Suggitt et al. 2018), a species that was present was very likely observed and non-recorded species are likely to be

true absences. Second, we include a further spatial autocorrelation control utilising the function 'pcnm' within the R package 'vegan'. Finally, our calculations of distribution change using the Frescalo algorithm were compared against the robust but more conservative (according to Isaac et al. 2014) Telfer metric, showing good agreement (Figure S2), which lends further support to the reliability of our more advanced, Frescalo-derived metrics.

In further response to this comment, we conducted an additional analysis to test if extirpations were more likely in areas with low recording effort. To do this, for each species, we calculated mean recording effort in hectads with extirpations and mean recording effort in hectads with persistences. Comparing the two, if a large proportion of species suffered from recording bias, then one would expect recording effort to be lower in hectads with extirpations (than in hectads with persistences). We can confirm that recording effort was lower in extirpated hectads for 61.2% of study species. In other words, for some ~39% of the species we analysed, there was arguably no (detectable) issue with recording effort, and for the remainder, by including recording effort as a control variable (as we did in all models), potential issues with spatial biases arising from recording patterns are accounted for.

In addition, it would be helpful to discuss any issues that arise when comparing across these time periods that also vary considerably in the time frames. I think the latter is probably less important for within taxon comparisons, but it does influence the general interpretation of the estimates and, given that sampling time frames vary across taxa, it makes direct comparisons across taxa challenging.

Response from the author team: In response to Reviewer 3's comment regarding possible phylogenetic effects on the outcomes of the taxon-level modelling (see below), we have now deprecated these results in favour of a wholly species-level approach, which reduces the emphasis on cross-taxon comparisons (although we would highlight in response to this comment that results were largely consistent across taxa, and between the taxon-level and species-level approaches).

We have also adjusted the wording that accompanies the description of the time periods to more clearly outline their justification: 'Although these differ somewhat from the windows for the land-use change data- which are based on the only available data at the national level prior to large-scale grassland loss and agricultural intensification¹⁵- they were chosen to best represent the exposure of each taxon to climate change, and they reduced the possibility of important climate shifts between the historical and modern recording periods going uncaptured by the climate change data we used to represent them'.

Reviewer #3 (Remarks to the Author):

An interesting paper using an exciting new dataset on land use change (which does look amazing) to look at how climate change and land use change might impact persistence in 4 groups.

Response from the author team: Many thanks for your positive comments on our manuscript.

So while I like this idea, I found the paper quite hard work and have lots of questions and queries, some pretty fundamental to the conclusions you draw. Apologies if I have the wrong end of the stick in places.

Line 29. It seems odd to say that evidence for additive or interactive effects is lacking – quite a few recent papers have explored and shown these effects – in vertebrates and insects etc...

Response from the author team: We agree- we have adjusted the language here to suggest the evidence is 'limited', rather than 'lacking'. (We go on to explain what the limitations are in the text itself).

Line 33 You say, 'to distribution changes' but you only model persistence.

Response from the author team: We do include a measure of overall distribution change in the analyses supporting Figure 3. We have tightened the wording in the results sentences of the abstract to make this more explicit ('and these effects were reflected in both contractions and expansions')- we hope this helps to improve the clarity here, but we would of course be open to further changes (in the abstract and elsewhere) should the review team deem them necessary.

Line 37. When you say, 'interactive effects were rare, being detected in less than 1 in 5 species.' An occurrence of 20% sounds high to me and of biological relevance. Rephrase/explain.

Response from the author team: As originally written, the first clause of this sentence was intended to reflect the gist of the paragraph in which we analyse the difference between models with and without the interaction term (results section, and Table S6). But in response to this comment, we have introduced more detail into the abstract to make the relative weakness of the interaction effects more explicit. Revised text now reads: 'Interactive effects were relatively rare, being detected in less than 1 in 5 species, and their overall effect on extinction risk was often weak'.

Line 37. When you say, 'Such individualistic responses emphasise the importance of including species-level information in policies targeting biodiversity and climate adaptation.' Spell out how you suggest that is done. It reads as a throw-away comment.

Response from the author team: Our newly-expanded discussion section more directly spells out how we see the role of species-level information as being important, alongside a number of other additional conclusions we would suggest that the reader can draw from our results.

Space is severely constrained in the abstract though, and we would argue that the point about individualistic responses necessitating the inclusion of species-level information in biodiversity or climate adaptation policies as being the most worthwhile of the points we could make here.

Line 52. Agree that efforts have been limited, there are big data problems, but why not reference here the relevant work that has been published?

Response from the author team: We have bolstered the referencing here to make the relevant work clear, and have included more such work in the discussion (when setting out how we feel our manuscript sits within it). We have also added more qualifiers to the sentence that outlines why this manuscript makes a novel contribution to the field. These sentences now read as follows: "How these effects combine⁶ or interact^{7,8} can define biological responses to environmental change. However, a lack of spatial and temporal coverage of gridded datasets quantifying these drivers- particularly so in the case of land-use change – has limited efforts to establish both their prevalence and their long-term effects (> 50 years) across a range of taxa".

Line 97. Interesting to see extirpations were lower where land conversion was also high, would you not predict the reverse? There's lot of evidence to show that the land use changes and the conversions you describe were associated with specific and general abundance and distributional declines. Are you testing particular hypotheses here and making predictions of effects? That might help structure the paper and help the reader?

Response from the author team: As Reviewer 3 is correct to point out, we didn't make explicit predictions on what the overall effects of land-use change and climate change might have been for each taxon. But to make our line of inquiry clearer, and also to address the L52 comment, we have introduced more previous work on this topic, including its typical findings on land-use change responses, and frame the MS (particularly the biodiversity analyses) in these terms.

Line 53. The new land use change map looks extremely useful and is carefully constructed. Amazing. I love the new maps on their own.

Response from the author team: Thank you for your positive comments on the new dataset.

Line 73. I like figure 1 but would like 1F to show the separate mapped changes in each of the 5 different land use on GB maps. Their pattern of spatial change would be extremely useful to visual and understand.

Response from the author team: Thanks for this suggestion- we have included the 5 additional panels in a revised Figure 1 as per your request.

Line 79. Here you say you excluded species with 100 or fewer records but for which period? Later you say the first period, which I query, because that excludes any natural colonists? Please clarify.

Line 83. If this paper only relates to persistence, then change the title.

Line 84. What about the opposite case of absence in the first period but presence in the second? Is that considered?

Line 287. I can see why you limited your analyses to species with more than 100 unique 10 × 10 km grid square records in the first time period, but why not also allow the same for the second period? Your methods seem to exclude colonising species, which seem highly relevant.

Response from the author team (to comments on lines 79-84 and 287): For the analyses that support Figure 2 (panels c and d), and those that determined species' response types in Figure 3, we excluded species with 100 or fewer unique records in the first time period, and we have now made this explicit in the main text. Revised text now reads: 'we excluded any species that was recorded in 100 unique 10 × 10 km grid squares or fewer in the first time period'.

Colonisations (absence in TP1 but presence in TP2) did form part of the overall distribution change measure included as the y-axis for Figure 3. We feel that this inclusion- and the demonstration that the changes we identified have a demonstrable effect on overall range shifts- warrant our use of 'species' range changes' in the title, and our use of the term 'distribution changes' in the abstract (see also our change in response to Reviewer 3's L37 comment above). But we are open to a further tightening of the language to contractions/retractions if the review team feels it is likely to aid the reader's understanding.

We would also point out here that our exclusion of colonisations from the statistical models (in Figure 2) is a further means of (conservatively) addressing the possible effects of recorder effort increases over time (see also Reviewer 2's comment on this), and the potential for false colonisations that could have arisen. (Note that Frescalo's national distribution change metrics that we adopt in Figure 3- y-axis, also replicated within Telfer- have been shown to be robust to this).

Line 102. The other piece of basic information relevant here, but missing from what I can see, would be a tabulation of the presence and absence by each taxonomic group. I am interested to understand the patterns of distributional changes in these taxa, that then sit behind the correlations. These changes are hinted at in the paper but unclear.

Response from the author team: Thanks for this suggestion. We have now included a table of the mean rates of persistence per species in the Supplementary Materials.

Line 114. Is there a word missing here?

Response from the author team: Yes- we have added the word 'of' here. Thanks for spotting this.

Line 152. You say, 'Taken in sum, our results are compatible with the highly variable, yet overall neutral trends in biodiversity at the local and landscape scales across the globe' but how so? Expand and explain. You aren't measuring biodiversity change per se, just range not abundance, lots of caveats in using unstructured presence data, attempting to correct for sampling effort, just four taxa etc. I suggest you delete this sentence and such speculative comparisons. It adds confusion.

Response from the author team: Agreed- changed as suggested.

Line 168. Is it safe to say, 'our results suggest that climate change has been a more important driver of population persistence over this period than land-use change', first given the data quality and just four taxa groups, plus the methods used, second, the inherent difficulty in quantifying the two in an equivalent way in relation to species responses? They are apples and oranges. A small change in land use, imperceptible when considering such broad categories of land use, change might determine the fate of a species. The extent of a land type may have no bearing on the intensity of land use within in - and the latter may be critical to species. I suggest you are more circumspect here and cautious and include some discussion of the real caveats here.

Response from the author team: We agree with the need for caution when comparing effects of climate and land use change, and have removed this particular phrasing from the general discussion as a result. We have also substantially expanded the discussion of our findings more generally around this point, and have included more caveats and possible explanations for what we found. We hope you agree that the revised material is more reflective of our results, and what can cautiously and robustly be concluded from them.

Line 173. Rather underwhelming and quite misleading to say, 'Management changes within land use categories, such as agricultural intensification, may also have been detrimental in recent decades.' with no reference to a body of research that shows such effects in a range of different ways. The emphasis on 'may' here but the whole paragraph feels unbalanced.

Response from the author team: We agree- we have bolstered this section in response to this comment (and that of Reviewer 1). Sentence now reads: 'Management changes within land use categories (as such beyond our scope here), such as agricultural intensification, have also been shown to be detrimental in recent decades (e.g. for birds²⁸), and could surpass land conversion as stronger drivers of change in regions (such as Great Britain) where land-use configurations are now largely stable'.

Line 177. Add references to the work showing those effects.

Response from the author team: We have made the wording more specific here, and added a couple of references. Sentence now reads: 'Climate change is increasingly being shown to be an important driver of biodiversity change^{23,24}.'

Line 186. The final sentence here, 'These factors,' is again a somewhat through-away comment. Spell out what is intended in plain language and be more specific in your recommendation.

Response from the author team: We have clarified the language here, but we would argue that the findings necessitate a somewhat equivocal conclusion. The revised sentence reads: 'These factors, alongside the tendency for species' responses to global change to be highly individual, mean that the ultimate outcomes for many reorganized populations and ecosystems remain unclear'.

Line 307. Why model taxon-level responses to land cover and climate change statistically? That is, why fit separate models for each taxa? Why not one model with taxa as a random or fixed effect? The thing you are interested in is the effects of climate warming and land conversion on species' range change – not testing differences between taxonomic groups etc.

Line 307. The models need to incorporate correction for phylogenetic similarity or at least test that such effects are not biasing you results.

Line 333. Why now embark on individual overlapping species-level modelling of the same effects?

Line 102. The individual species models now seem to duplicate the results presented in the previous paragraph in a slightly more jumbled fashion. I see these feed into fig 2b and 2b Is this better presented as supplementary analyses?

Line 575. Fig 2 – no correction for phylogenetic effects here.

Line 584. Fig 3 – again no correction for phylogenetic effects here, which is problematic as the counts of numbers of species showing effects is being presented, regardless of their relatedness.

Response from the author team: In our study, we consider changes in species occurrence over time. As such, our data is structured according to how each observed species across our taxonomic groups has been recorded in grid squares across Britain. We worked closely with taxonomic experts to assemble the lists of grid square records (and datasets) for each species we analyse here, and we can confirm that each of the 'species' we analysed has just cause to be considered as such (and we go with 'atlas', or standard recording definitions in almost all cases). The care required in this process is reflected in the co-authorship of the manuscript by data scientists for all four taxa. Some of the species we analysed will of course be more (or less) related to the others, but we do not think that this makes their responses to environmental change more or less important than other species.

We do follow work that applies phylogenetic corrections in the literature, for example when analysing how (for example) functional traits affect species' occurrences or turnover, to ensure that any patterns

observed in the data (e.g. large-bodied species more likely to exhibit a specific response) are due to the trait in question's role in responding to the environment, rather than that trait merely reflecting the species' evolutionary history (although there are voices arguing that we should accept trait-related responses regardless of how, where and when these traits developed – e.g. Westoby et al. 1995, *Journal of Ecology*; de Bello et al. 2015, *Folia Geobotanica*). But we do not consider such a correction to be necessary here, where we consider each species as a equal unit of biodiversity, regardless of its evolutionary history prior to being observed in Britain during our first time period.

Nonetheless, and to reduce the potential confusion that might arise from conducting our analyses at two levels (taxon level and species level), we have deprecated the taxon-level results (those most likely to be affected by phylogenetic relatedness) and moved them to the Supplementary Materials. We now only make reference to them- and specifically the large species-level random effect we found in all the taxa- as a justification for conducting our subsequent analyses at the species level. However, we are convinced that all species should be given equal weighting in our summaries of species-level change over time.

Thank you for highlighting these issues- we hope the text is much clearer as a result of this change.

Line 336. Explain in more specific detail how perfect prediction occurs in this case and how it affects your analysis of biodiversity change.

Response from the author team: We have elaborated on the text here. New text now reads: 'Where cases of complete separation occurred (n = 39 species)- where persistences/extirpations formed two non-overlapping cohorts along the observed values of a predictor- the relevant species was removed from our analyses, both at the species level and at taxon level. These 39 species tended to be common widespread species with very high rates of persistence, and as such were unlikely to be responsive to environmental change. These species were also excluded from the total numbers of species in each taxon quoted in the 'Biodiversity change' section above'.

Line 365. You report that higher rates of land conversion may also have been subject to higher recorder effort – that would explain artefactually why you find, rather surprisingly, positive relationships between persistence and land conversion (Fig 2). That is a major worry.

Response from the author team: With apologies for the imprecise language, this text refers to the tests we were about to conduct, rather than reporting the findings of the tests we did conduct. We have amended this to read: 'To test if squares subject to higher rates of land conversion were also subject to higher recorder effort...'

Line 370. How does the correlation between mean annual temperature and temperature change influence your findings? You describe this as a weak correlation ($r = 0.56$, $p < 0.0001$) - but many would kill for such a correlation.

Response from the author team: Thanks- we have added discussion of this in the concluding material of the main text, which reads: 'We would also point out that, unlike global patterns of higher temperature change in cooler regions, Great Britain has in fact tended to experience faster rates of warming in areas with warmer mean temperatures (Pearson's $r = 0.56$, shared variance of 31%; see Methods). This could mean that a component of the positive responses to climate change may in fact be positive responses to higher mean temperatures in the environment. This positive correlation also makes it unlikely that our approach has mischaracterized species' responses to (warmer/cooler) thermal environments'.

RD Gregory

REVIEWERS' COMMENTS

Reviewer #2 (Remarks to the Author):

I have one general remaining suggestion: it would be useful in the Discussion to briefly provide context for these results beyond Great Britain. Currently, there is no discussion of how this work fits in a global context. Yet it strikes me that Great Britain has had considerable intensive land use over much longer periods than many areas in the world (and prior to this long time series), it is an island, and it is at a very northern latitude. It would be helpful to add some discussion for how this work is relevant elsewhere, given these issues.

I have some specific suggestions that I hope will improve the manuscript.

Line 78. Around here, it would be helpful for the authors to explain what is meant by 'land conversion' so that readers can get a better sense of these results without needing to look at the details in the methods.

Line 98. I would change 'survival' to 'persistence' to be consistent with the terms used in the preceding sentences.

Line 123. Consider changing 'predicted' to 'estimated', since the authors emphasized in their response letter that the point here is not on predictions.

Line 148. Please be more specific here what the '10% change' is—increase in semi-natural grasslands?

Reviewer #3 (Remarks to the Author):

The authors have responded to the review comments in a positive and helpful fashion and the revised manuscripts feels much improved. I only have a small number of quite minor comments although I would like to see the specialist species modelled in a slightly different way, which might have little effect, but would be good to bottom out.

Specific comments

Line 98. Perhaps change 'survival' to 'persistence' in 'proportion of variation in survival explained by this effect'

Line 113, 115 & 119 – I think you are referring to Figs 2 c & d here – not b & c?

Line 142. I don't really follow why having identified 'specialist' species you substitute the generic 'land conversion' variable for relative change in the specific habitat type each specialist was associated with. I would have just used the generic land conversion variable and see if these species were more sensitive to general change. The restriction feels too restrictive. I guess each tells you something different.

193. Change 'unlikely to beneficial to species' to 'unlikely to be beneficial to species'

RD Gregory

REVIEWERS' COMMENTS

[No comments from Reviewer #1].

Reviewer #2 (Remarks to the Author)

I have one general remaining suggestion: it would be useful in the Discussion to briefly provide context for these results beyond Great Britain. Currently, there is no discussion of how this work fits in a global context. Yet it strikes me that Great Britain has had considerable intensive land use over much longer periods than many areas in the world (and prior to this long time series), it is an island, and it is at a very northern latitude. It would be helpful to add some discussion for how this work is relevant elsewhere, given these issues.

Authors: Thanks for this suggestion. We agree, and we have added more context for the results beyond Great Britain.

I have some specific suggestions that I hope will improve the manuscript.

Line 78. Around here, it would be helpful for the authors to explain what is meant by 'land conversion' so that readers can get a better sense of these results without needing to look at the details in the methods.

Authors: Thanks, we have now promoted the definition further up. Sentence now reads: 'We used the new land-use change dataset and the existing climate change dataset to look for single, combined, and interactive effects of temperature warming (Fig. 2a) and land conversion (which we define as the proportion of 25 x 25 m pixels within a 10 x 10 km grid square that had changed land-use class over time, Fig. 2b) on species' distribution changes'.

Line 98. I would change 'survival' to 'persistence' to be consistent with the terms used in the preceding sentences.

Authors: Changed as requested.

Line 123. Consider changing 'predicted' to 'estimated', since the authors emphasized in their response letter that the point here is not on predictions.

Authors: Changed as requested.

Line 148. Please be more specific here what the '10% change' is—increase in semi-natural grasslands?

Authors: We have made this sentence more specific, and it now reads: 'Grassland specialists appeared more sensitive, with a +10% change in semi-natural grassland habitat (more a case of retention rather than increase; Fig. 1g) resulting in a median 1.95% increase in the persistence probability'.

Reviewer #3 (Remarks to the Author)

The authors have responded to the review comments in a positive and helpful fashion and the revised manuscripts feels much improved. I only have a small number of quite minor comments although I would like to see the specialist species modelled in a slightly different way, which might have little effect, but would be good to bottom out.

Authors: Many thanks for these comments, see our responses to these specific queries below.

Specific comments

Line 98. Perhaps change 'survival' to 'persistence' in 'proportion of variation in survival explained by this effect'.

Authors: Changed as requested.

Line 113, 115 & 119 – I think you are referring to Figs 2 c & d here – not b & c?

Authors: Very well spotted, changed as requested, many thanks!

Line 142. I don't really follow why having identified 'specialist' species you substitute the generic 'land conversion' variable for relative change in the specific habitat type each specialist was associated with. I would have just used the generic land conversion variable and see if these species were more sensitive to general change. The restriction feels too restrictive. I guess each tells you something different.

Authors: Our decision to test for an effect of change in the specific type of land use (vs deploying the more generic land conversion variable) was in response to Reviewer #2's comment to 'explicitly probe the types of land change' in the previous round of reviews. We agree that the two ways of doing it would tell you something different, although we'd highlight that any changes in types that the target species aren't sensitive to- bundled within the land conversion estimates for each grid square- would simply add noise to their response signals (such as they are).

If the analyses of specialists we did conduct (in Figs S3 and S4) returned particularly strong or materially different effects to those we report in the main text, then there might be a case for seeing if they remained strong/different when using the less calibrated, land conversion variable. But given that this is only likely to generate weaker associations than those we already report (as it contains more noise) then we would prefer to retain the (more tuned) analyses that we already have.

193. Change 'unlikely to beneficial to species' to 'unlikely to be beneficial to species'

Authors: Again well spotted, changed as requested.

RD Gregory